# Transcriptome and Metabolome Analyses Reveal Mechanisms Underlying the Response of Quinoa Seedlings to Nitrogen Fertilizers

**DOI:** 10.3390/ijms241411580

**Published:** 2023-07-18

**Authors:** Hanxue Li, Qianchao Wang, Tingzhi Huang, Junna Liu, Ping Zhang, Li Li, Heng Xie, Hongxin Wang, Chenghong Liu, Peng Qin

**Affiliations:** 1College of Agronomy and Biotechnology, Yunnan Agricultural University, Kunming 650201, China; 2021210172@stu.ynau.edu.cn (H.L.); 2020110028@stu.ynau.edu.cn (Q.W.); 2020240162@stu.ynau.edu.cn (T.H.); 2021110026@stu.ynau.edu.cn (J.L.); 2021110031@stu.ynau.edu.cn (P.Z.); 2019210130@stu.ynau.edu.cn (L.L.); 2020210159@stu.ynau.edu.cn (H.X.); 2021240157@stu.ynau.edu.cn (H.W.); 2Shanghai Key Laboratory of Agricultural Genetics and Breeding, Biotech Research Institute, Shanghai Academy of Agricultural Sciences, Shanghai 201106, China

**Keywords:** quinoa seeding, nitrogen, metabolomics, transcriptomics

## Abstract

Quinoa (*Chenopodium quinoa* Willd.) is a dicotyledonous annual amaranth herb that belongs to the family Chenopodiaceae. Quinoa can be cultivated across a wide range of climatic conditions. With regard to its cultivation, nitrogen-based fertilizers have a demonstrable effect on the growth and development of quinoa. How crops respond to the application of nitrogen affects grain quality and yield. Therefore, to explore the regulatory mechanisms that underlie the responses of quinoa seedlings to the application of nitrogen, we selected two varieties (i.e., Dianli-1299 and Dianli-71) of quinoa seedlings and analyzed them using metabolomic and transcriptomic techniques. Specifically, we studied the mechanisms underlying the responses of quinoa seedlings to varying concentrations of nitrogen by analyzing the dynamics of metabolites and genes involved in arginine biosynthesis; carbon fixation; and alanine, aspartate, and glutamate biosynthetic pathways. Overall, we found that differentially expressed genes (DEGs) and differentially expressed metabolites (DEMs) of quinoa are affected by the concentration of nitrogen. We detected 1057 metabolites, and 29,012 genes were annotated for the KEGG. We also found that 15 DEMs and 8 DEGs were key determinants of the differences observed in quinoa seedlings under different nitrogen concentrations. These contribute toward a deeper understanding of the metabolic processes of plants under different nitrogen treatments and provide a theoretical basis for improving the nitrogen use efficiency (NUE) of quinoa.

## 1. Introduction

Quinoa is an annual dicotyledonous plant of the family Amaranthaceae, originating from the Andes in South America [1]. Quinoa has been classified as a “golden grain”, as it is the only plant recognized by the International Food and Agriculture Organization (FAO) as having a nutritional profile that allows it to meet all the basic nutritional needs of humans [2]. Quinoa is rich in protein [3], contains most natural amino acids, and consists of numerous mineral elements, vitamins, and alkaloids [4,5,6,7,8,9,10]. As a whole grain, quinoa can reduce the risk of cardiovascular disease and support digestive health [11,12,13]. It is also rich in vitamin E, which is an important antioxidant that effectively inhibits the oxidation of oil and counteracts lipid peroxidation damage of the cell membrane, yielding anti-aging effects [14]. In addition, within the context of agriculture, quinoa has strong resistance to environment-linked stress.

Nitrogen plays an important role in terms of the growth and development of a variety of agricultural products, greatly improving resultant yields [15,16,17]. However, the excessive use of nitrogen fertilizers increases the cost of agricultural production and poses a serious threat to soil health and the ecological integrity of the surrounding environment and the organisms that inhabit it [18]. Nitrogen use efficiency (NUE) is an index used to benchmark the use of nitrogen fertilizer [19]. It refers to the uptake and use of nitrogen by plants and allows for the quantification of the benefits of nitrogen across various developmental processes of plants. NUE is particularly important within the context of sustainably maintaining high crop yields. The use of nitrogen fertilizers, which is rooted in science, and the improvement of NUE are internationally recognized as key research topics of the 21st century [20]. Recently, researchers have linked improved yields with the improvements in the efficiency at which nitrogen-based fertilizers are processed by plants, thus significantly reducing the amount of fertilizer that has to be used [21]. Fradgley et al. showed that the NUE of wheat grown under high-nitrogen (HN) treatments is much lower than that grown under low-nitrogen (LN) treatments [22]. In most terrestrial habitats, nitrogen is one of the factors that limit the growth of plants; this is because the synthesis of nucleic acids and proteins requires large quantities of nitrogen, and the availability of nitrogen directly determines the composition and content of storage proteins [23,24]. An excess amount of nitrogen results in large amounts of carbohydrates being used to synthesize substances such as proteins and chlorophyll; this results in plant cell walls losing a lot of cellulose and pectin, yielding cells that are large and thin-walled. Previous studies have shown that the accumulation of amino acids and anthocyanins plays an important role in plant response to nitrogen stress [25,26].

Metabolomics not only enables the characterization of endogenous substances in organisms but also allows for the characterization of the mechanisms that shape the dynamics of these substances [27]. On the other hand, transcriptomics can allow for the elucidation of gene expression across a wide range of environmental stressors; specifically, this is accomplished through sequencing the RNA of all transcripts in plants subjected to a specific stressor [28]. The seedling stage is the transitional stage from heterotrophic to autotrophic and is a key period for ensuring high crop yield. If the growth status of the seedling stage is restricted, it will have a negative impact on yield. The mechanisms by which quinoa seedlings respond to phosphorus fertilizer and potassium fertilizer dosage have previously been reported, and Wang et al. jointly analyzed the transcriptome and metabolome and found that quinoa seedlings respond to different levels of phosphorus fertilizer by regulating some metabolites and genes in the glycolysis, glyceride, and glycerol phospholipid pathways [29]. Huang et al. combined transcriptome and metabolome analyses and found that quinoa seedlings responded to different levels of potassium fertilizer through some metabolites and genes in the photosynthetic pathway [30]. However, to the best of our knowledge, the mechanisms underlying the responses of quinoa seedlings to varying concentrations of nitrogen have not been investigated. This experiment used urea as nitrogen fertilizer. Previous studies have reported that quinoa, as well as sugar beet and spinach, which are also plants in the amaranth family, use urea as a source of nitrogen fertilizer. Previous studies found that the application of urea results in better growth compared to the application of ammonium nitrate [31,32,33,34]. Therefore, we combined transcriptomic and metabolomic approaches to characterize these mechanisms. This study aims to provide fine-scale insights into the differential expression of metabolites and genes associated with nitrogen use efficiency in quinoa. Additionally, this study provides a theoretical basis for efficient breeding, high-yield cultivation, and sustainable use of nitrogen-based fertilizers within the context of quinoa cultivation.

## 2. Results

### 2.1. Changes in Agronomic Characteristics of Quinoa Seedlings under Different Treatments

Quinoa seedlings grew optimally at 112.5 kg/hm^2^ and 0 kg/hm^2^ of CH_4_N_2_O; in contrast, they became stiff and short at 337.5 kg/hm^2^ of CH_4_N_2_O. The morphology of quinoa seedlings changed significantly after 30 d of fertilization treatment, and quinoa seedlings were in the six-leaf stage. Specifically, the height of red quinoa under HN was significantly lower than that under CK and LN, and that of LN was marginally lower than that of CK. The plant height of white quinoa was roughly similar between the CK group and the HN and LN groups (Figure 1a,b). These results showed that as the nitrogen content increased, the height of quinoa increased; however, beyond a certain concentration of nitrogen, the plant growth is affected (Figure 1c). The leaf area of red quinoa under HN and LN was lower than that under CK, whereas the leaf area of white quinoa was slightly lower in the LN and CK groups than in the HN group, with no significant differences (Figure 1d).

### 2.2. Qualitative and Quantitative Analysis of Metabolites in Quinoa Seedlings under Different Treatments

The seedlings after 30 d of fertilization and in the six-leaf stage of quinoa were divided into six groups (two cultivars and three treatments), each with three biological replicates; 1057 metabolites were detected across the 18 samples. Among them, there were 98 amino acids and their derivatives, 166 phenolic acids, 69 nucleotides and their derivatives, 172 flavonoids, 11 quinones, 24 lignin and coumarins, 9 tannins, 78 alkaloids, 39 terpenoids, 94 organic acids, and 177 lipids (Appendix A). From the total ion flow diagram of the MS analysis of the different quality control samples, it is apparent that the ion peaks of identical substances in duplicate samples overlap (Appendix A), indicating that the experimental process was stable and that the results are reliable. Correlation analysis between samples showed that there was a strong correlation between the results of different nitrogen treatments (Appendix A). The CV diagram (Figure 2a) shows that the proportion of QC sample materials with small data dispersion and CV value less than 0.5 is higher than 85%, indicating that the experimental data are relatively stable. In addition, principal component analysis could be used to determine the variability of red and white quinoa seedlings among and within LN, CK, and HN groups. The PCA score chart of all samples (Figure 2b) shows that the original data obtained from UPLC-MS/MS analysis were largely shaped by PC1 and PC2. The contribution rate of PC1 and PC2 was 37.16% and 15.76%, respectively. This indicated that both principal components reflected the main feature information of the test samples. The six groups of samples showed noticeable separation on the two-dimensional graph, especially with regard to the red quinoa samples, indicating that the data processing of each sample was reliable with significant differences among the samples. The Venn diagram (Figure 2c and Appendix A) shows the common and unique DEMs among the four comparison groups, and the cluster heat diagram of differential metabolites (Figure 2d) shows the difference in DEMs between quinoa seedlings under different nitrogen treatments.

The difference in multiple histograms (Figure 3a–f) and the differential metabolite VIP value diagram (Appendix A) show that the significant DEMs among the groups were phenolic acids, flavonoids, lipids, amino acids and derivatives, and organic acids. K-means clustering analysis was conducted after the z-score standardization of the relative content of the DEMs (Figure 3g). Using K-means cluster analysis, DEMs were divided into 12 clusters. In the comparison between the CK and HN groups, clusters 5, 9, 10, and 12 significantly increased and cluster 6 decreased significantly in the HN group. In the comparison between the CK and LN groups, clusters 5, 7, 9, and 10 significantly decreased in the LN group.

### 2.3. Screening of DEMs and Enrichment Analysis of the KEGG Pathways

OPLS-DA was used to analyze the differences between sample groups and screen for differential metabolites. DEMs were screened based on the VIP and difference multiple values (VIP > 1; FC > 2 or FC < 0.5) of the OPLS-DA model. After converting to the log2 scale, there were 74 DEMs in the different multiples of red quinoa under LN and CK. Of these, 13 were upregulated while 61 were downregulated. Of the 58 differential metabolites in white quinoa under LN and CK, 26 were upregulated while 32 were downregulated. There were 11 common DEMs between the two cultivars (Appendix A). There were 274 differential metabolites in red quinoa under HN and CK, of which 133 were upregulated while 141 were downregulated. There were 52 differential metabolites in white quinoa under HN and CK, of which 41 were upregulated while 9 were downregulated. There were 22 common DEMs between the two cultivars (Appendix A). There were 337 differential metabolites in red quinoa under LN and HN, of which 197 were upregulated while 140 were downregulated. There were 77 differential metabolites in white quinoa under LN and HN, of which 63 were upregulated while 14 were downregulated. There were 29 common DEMs between the two cultivars (Appendix A). The volcanic map of the above four combinations clearly shows the abovementioned trend (Appendix A). KEGG is the main public database related to pathways, and it helps in studying genes and metabolites as a whole network. The screening standard for obtaining KEGG pathways of different metabolites was *p* < 0.05. There were 4 significant pathways in LN vs. CK, 7 in HN vs. CK, and 13 in LN vs. HN of red quinoa. There were two significant pathways in LN vs. CK, one in HN vs. CK, and three in LN vs. HN of white quinoa. The enrichment degree of differential metabolites in the pathways can be seen from the KEGG enrichment map (Figure 4a–f). The DEMs detected in all samples were mainly enriched in phenylpropane biosynthesis, flavonoid biosynthesis, starch and sucrose biosynthesis, and arginine biosynthesis.

### 2.4. Transcriptome Analysis of Quinoa Seedlings under Differential Nitrogen Application

Fertilization was conducted at the two-leaf stage of quinoa, and samples were taken after 30 days of fertilization. A total of 18 samples from red and white quinoa strains under three different nitrogen treatments (three biological replicates) were sequenced for transcriptome analysis. A total of 134.27 Gb of clean data was generated. The clean data of each sample reached 6 Gb, the GC content was 42.77–47.2%, and the base percentage of Q30 (sequencing error rate < 0.1%) was 91% or more. Sequences were compared between clean reads after QC. If the reference genome assembly is relatively complete and the tested species are consistent with the reference genome, the proportion of sequencing reads generated in the experiment successfully aligned with the genome will be higher than 70%, and the comparison efficiency of this sequencing is higher than 80% (Appendix A). From the transcriptome data above, it was apparent that the sequences were of a high quality, which provided a basis for the subsequent DEG analysis. The density map showed that the gene abundance in the 18 samples changed with the amount of expression, clearly showing that the gene expression (FPKM) in samples was concentrated between log10^−2^ and log10^4^ (Figure 5a). Correlation analysis was conducted among the duplicates of each group. The correlation heat map (Figure 5b) showed that the correlation between corresponding groups of samples was >0.8 except for that of W3-6, which was <0.8. W3-6 was removed when analyzing the data to ensure the reliability of the results. The comprehensive correlation heat map and box graph (Figure 5c) showed that the biological repeatability of transcriptomics of quinoa samples of the two varieties was good and that there were differences between them. Functional annotation of genes was detected in this experiment in NR, Swiss Prot, GO, COG, KOG, Pfam, KEGG, and other databases. The results showed that the number of genes annotated in the NR database was 52,783; the number of genes annotated in the Swiss Prot database was 34,021; the number of genes annotated in the GO database was 40,842; the number of genes annotated in the KOG database was 48588; the number of genes annotated in the Pfam database was 44,256; and 29,012 genes were identified in the KEGG database, involving 139 pathways.

Using |log2Fold Change| ≥ 1 and FDR < 0.05 as screening conditions, we screened DEGs among different samples and used DESeq2 for further analysis of DEGs. In red quinoa, the LN vs. CK comparison group revealed 513 DEGs, of which 239 were downregulated while 274 were upregulated. In the HN vs. CK comparison group, 338 DEGs were identified, of which 79 were downregulated while 259 were upregulated. In the LN vs. HN comparison group, 653 DEGs were identified, of which 316 were downregulated while 337 were upregulated. (Appendix A). In white quinoa, the LN vs. CK comparison group identified 29 DEGs, of which 20 were downregulated while 9 were upregulated. In the HN vs. CK comparison group, 11 DEGs were identified, of which 10 were downregulated while 1 was upregulated. In the LN vs. HN comparison group, 23 DEGs were identified, of which 22 were downregulated while 1 was upregulated. (Appendix A). The results above show that the number of DEGs in quinoa seedlings under LN and HN conditions was significantly higher than that in the CK group, indicating that these DEGs were important in the response of quinoa seedlings to nitrogen fertilizers (Appendix A).

We conducted KEGG pathway analysis and GO enrichment analysis for all DEGs, and DEGs in a range of metabolic pathways were mapped to the KEGG database. Through the analysis, we found that differential genes were enriched in metabolic pathways related to nitrogen metabolism in the LN vs. CK, HN vs. CK, and LN vs. HN groups. The metabolic pathways associated with most DEGs were primarily linked to DNA replication, fatty acid metabolism, fatty acid biosynthesis, and flavonoid biosynthesis. In the LN vs. CK comparison group, 217 significantly enriched GO functions were obtained, including 28 cell components (CCs), 67 molecular functions (MFs), and 122 biological processes (BPs). In the HN vs. CK comparison group, 181 significantly enriched GO functions were obtained, including 18 CCs, 17 MFs, and 146 BPs. In the LN vs. HN comparison group, 104 significantly enriched GO functions were obtained, including 10 CCs, 67 MFs, and 27 BPs. It is apparent from the enrichment column diagram of differential GO (Figure 6a–c) that DEGs in the three comparative groups of red quinoa, LN vs. CK, HN vs. CK, and LN vs. HN, are mainly enriched in “biological processes” such as the fatty acid metabolic process, fatty acid biological process, organic acid transport, ion transmembrane transport, and responses to organic compounds. The transcription factors related to the response of quinoa seedlings to nitrogen fertilizer include FAR1, bHLH, MYB, B3, C2H2, AP2/ERF-ERF, NAC, and WRKY (Figure 5d).

### 2.5. Transcriptome and Metabolome Analysis of the Nitrogen Regulation Mechanism in Quinoa Seedlings

For a systematic and comprehensive analysis of the molecular mechanism underlying the regulation of nitrogen deficiency and high nitrogen in quinoa seedlings, we integrated both the metabolome and transcriptome data and simultaneously mapped DEMs and DEGs in the same group to the KEGG pathway and constructed a histogram according to the enrichment results of the DEMs and DEGs. In the LN vs. CK group, DEMs and DEGs were significantly enriched in the arginine biosynthesis and flavonoid biosynthesis pathways, while in the HN vs. CK group, DEMs and DEGs were significantly enriched in the starch and sucrose metabolism pathways (Appendix A). The DEMs and DEGs annotated to carbon fixation in photosynthetic organisms, arginine biosynthesis, alanine, aspartate, and glutamine metabolism in quinoa seedlings under different nitrogen treatments are shown in Figure 7.

In arginine biosynthesis, NOS1 was upregulated under HN treatment. Correlation analysis (Table 1) of DEGs and DEMs showed that NOS1 had a high positive correlation with four metabolites (PPC > 0.8), namely L-ornithine, L-glutamine, L-arginine, and N-α-acetyl-L-ornithine. Notably, this gene was not expressed under LN treatment, and these four metabolites were significantly downregulated under LN treatment. The above result shows that the urea cycle was inhibited under the LN treatment. In the metabolism of alanine, aspartate, and glutamate, under the LN condition, downregulated expression of GLUD1_2 was negatively correlated with 2-oxoglutarate (*p* > 0.8), and 2-oxoglutarate was upregulated, which led to the downregulation of oxaloacetate and ASS1 expression. For carbon fixation in photosynthetic organisms, phosphoenolpyruvate, dihydroxyacetone phosphate, 3-photosho-D-glyceric acid, D-erythrose-4-phosphate, ribulose-5-phosphate, D-fructose-6-phosphate, and D-fructose-1,6-biphosphate were downregulated under HN conditions, and phosphoenolpyruvate carboxylase (ppc) and pyruvate, orthophosphate dikinase (ppdK) were upregulated. Under LN treatments, only erythrose-4-phosphate was upregulated, whereas other substances were not significantly expressed, and the ribulose bisphosphate carboxylase large chain (rbcL) was downregulated.

In addition, we analyzed the FPKM expression degree of nitrogen-related genes in comparison with CK. In CK vs. LN, GLUD1_2 [EC:1.4.1.3] gene-*LOC110704768*, ASS1 [EC:6.3.4.5] gene-*LOC110703812*, and rbcL [EC:4.1.1.39] *novel 420* expression significantly decreased. ACY1 [EC3.5.1.14] *novel 7764*, ASNS [EC:6.3.5.4] gene-*LOC110720129*, gene-*LOC110693243*, ppdK [EC:2.7.9.1] gene-*LOC110719952*, and gene-*LOC110690041* were significantly upregulated. In CK vs. HN, NOS1 [EC:1.14.13.39] gene-*LOC110710546*, ppc [EC:4.1.1.31] gene-*LOC110720234*, ASNS [EC:6.3.5.4] gene-*LOC110720129*, gene-*LOC110693243*, ppdK [EC:2.7.9.1] gene-*LOC110719952*, and gene-*LOC110690041* were significantly upregulated.

### 2.6. RT-qPCR

To confirm the accuracy of the RNA-seq data, we selected validated genes in pathways closely related to nitrogen metabolism for the RT-qPCR analysis. The expression pattern detected via RT-qPCR correlated with the sequencing results, and the RT-qPCR results of six genes (gene-*LOC110689260*, gene-*LOC110694697*, gene-*LOC110703812*, gene-*LOC110704768*, gene-*LOC110710546*, and gene-*LOC110714529*) were consistent with RNA-seq data(Table 2). This showed that the transcriptome sequencing results were reliable (Figure 8a–f).

## 3. Discussion

Quinoa seeds are rich in nutrients, with an amino acid composition similar to that of milk, approaching the ideal protein balance recommended by the Food and Agriculture Organization (FAO). Additionally, quinoa is also rich in proteins, lipids, vitamins, and minerals. Therefore, quinoa is known as a “full nutrient crop” [35,36]. Nitrogen supply limits the growth and development of plants. However, how quinoa adapts to the low-nitrogen soil environment in the Southern Tablelands of Bolivia and how to use nitrogen efficiently need further research [37,38]. However, no studies have been conducted on the molecular patterns underlying the responses of quinoa seedlings to varying concentrations of nitrogen. In this study, the phenotypic changes of white quinoa under three treatments were not significant, while we found that red quinoa seedlings grew to a certain extent under HN conditions and then transitioned toward becoming stiff, whereas the CK group and LN group continued to grow. The growth rate of the LN group was marginally lower than that of the CK group, and the difference between the two groups was not significant. The HN group was structurally the shortest, and it had the relatively smallest leaf area, while the CK group was the tallest and had the relatively largest leaf area, indicating that different nitrogen treatments have a substantial impact on the growth of quinoa. Therefore, we used broad target metabolomics and transcriptomics to explore the mechanisms underlying the responses of quinoa seedlings to varying concentrations of nitrogen. Kaul et al. reported that quinoa is sensitive to nitrogen fertilizers. Although increasing nitrogen fertilizer has little impact on the harvest index, the nitrogen content of quinoa grains greatly increases [39]. Berti et al. found that the yield of quinoa increases with an extended application of nitrogen fertilizer [40]. When the amount of applied nitrogen exceeded the optimal amount, NUE began to decline, and the harvest index decreased significantly; these trends are congruent with those observed in this study. Research on other plants in the Amaranthaceae family, such as sugar beet, in response to nitrogen stress found that low-nitrogen treatment was more effective in inhibiting the growth of sugar beet seedlings than high-nitrogen treatment [41]. Wang et al. found that sugar yield also increased with the increase in nitrogen application, and after reaching a level, sugar yield decreased with the increase in nitrogen application [32]. When the nitrogen application rate of spinach reaches 360 kg/hm^2^, the plant height, leaf area, and yield all reach their maximum values [34]. Therefore, it is necessary to explore an appropriate nitrogen usage rate in the production of quinoa to maximize nitrogen utilization efficiency and obtain higher quinoa biomass.

Nitrogen is the most demanded essential macroelement in plants and the primary nutrient limiting productivity in several ecosystems. Under HN conditions, NUE (i.e., yield per unit of available nitrogen) is often low [42]. Arginine is one of the principal amino acids and is a harmless NH4+ repository in quinoa leaves. The ASS1 gene encodes arginine succinate synthase (ASS), which is one of the rate-limiting steps in catalyzing arginine biosynthesis [31]. Arginine metabolism is important in the distribution and circulation of nitrogen in plants, as most nitrogen is stored in arginine [43,44]. In our study, there was a significant difference in the expression of enzyme genes related to the arginine biosynthesis pathway under low- and high-nitrogen conditions. Under low-nitrogen treatment, ASS1 expression was downregulated and arginine content decreased. It is speculated that under nitrogen-deficient conditions, plants increase nitrogen utilization by reducing arginine synthesis. N-acetylcholine forms ornithine under the catalysis of aminoacylase (ACY1), and its downregulation under LN conditions causes the upregulated expression of ACY1. Given the negative correlation between ACY1 and L-citrulline, ornithine was upregulated under LN conditions, resulting in the overall downregulation of metabolites in the urea cycle. Notably, all these metabolites were upregulated under HN treatment, and this difference is attributed to a regulatory gene known as NOS1, indicating that NO is involved in the nitrogen metabolism pathway of quinoa. As a gas molecule, NO participates in many biological processes such as plant growth and development and stress response, and it forms a complex interactive regulatory network with signaling molecules such as plant hormones and reactive oxygen species to finely regulate various stages of plant growth and development to maintain the normal life activities of plants [45,46]. Nitric oxide synthase (NOS) is one of the most complex and largest enzymes known thus far. It has been widely studied in animals. At present, similar substances with NOS activity have been detected in some plants, which can convert L-arginine into L-citrulline [47,48,49]. Under HN treatment, the expression of the NOS1 gene led to the upregulation of metabolites related to the urea cycle, suggesting that NO participated in the nitrogen metabolism of quinoa through the urea cycle. In addition, the alanine, aspartate, and glutamine metabolisms also play an important role in the nitrogen metabolism of quinoa.

Glutamine and asparagine facilitate the transportation of nitrogen in plants. When plants extract internal ammonia in large quantities, the ammonia forms glutamine or asparagine to relieve the toxicity associated with free ammonia [50]. In this study, glutamine greatly accumulated under HN, indicating that glutamine plays an important role in regulating nitrogen metabolism in quinoa. This result in is agreement with previous studies on spinach response to nitrogen stress and sugarbeet response to nitrogen stress, which found that the glutamine transferase (GAT1) gene was strongly induced in the root system, and the activity of glutamine synthetase (GS) increased in response to nitrogen stress under low-nitrogen conditions [41,51]. It has also been found that exogenous plant hormones help plants resist nitrogen stress by improving GS activity [52]. Lyu et al. found that exogenous nitrogen regulates starch, sucrose, and amino acid metabolism and other related metabolic pathways and promotes the synthesis of asparagine and NO; these results are consistent with those of our study [53]. Mauceri et al. found that under nitrogen-deficient conditions, the levels of L-aspartic acid and L-asparagine in eggplants increased, and those of particle-bound starch synthase (WAXY) and endonuclease decreased. However, in the present study, we did not observe a significant difference in L-aspartic acid and L-asparagine content [54]. GLUD1_2 is a mitochondrial enzyme that not only catalyzes the synthesis of glutamate from 2-oxoglutarate and ammonia but also catalyzes the reversible oxidative deamination of glutamate into 2-oxoglutarate, which presents a carbon skeleton for the TCA cycle and has a central function in ammonia metabolism [55,56]. Zhang et al. found that Chinese cabbage genotypes with high NUE strongly expressed genes related to auxin biosynthesis and glutamate dehydrogenase [57]. Ou et al. found that some key metabolites in the TCA cycle are sensitive to nitrogen [58]. In this study, under LN conditions, oxaloacetate was downregulated while 2-oxoglutarate was upregulated in the TCA cycle. This verified the results of Qu et al. and Zamani-Nour et al., who found that oxaloacetate outputs glutamate to the cytoplasm by supplying 2-oxoglutarate for the GS/GOGAT reaction [59]. Joshi et al. found that spinach significantly upregulated phosphoenolpyruvate carboxyl kinase (PEPCK) related to carbon metabolism under HN conditions, and phosphoenolpyruvate carboxyl kinase (PEPCK) catalyzed oxaloacetate to produce phosphoenol pyruvate (PEP) [51]. Phosphoenolpyruvate carboxylase is known to be upregulated under HN conditions, which is consistent with the results of this study, indicating that phosphoenolpyruvate carboxylase is involved in protecting plants against stress under high levels of nitrogen [59]. Multiple studies have found that transcriptional factor subsets composed of bHLH, MYB, WRKY, and AP2/ERF family members are strongly expressed in response to nitrogen disturbances, which is consistent with the results of this study [42,51,53,60]. In conclusion, the response mechanism of quinoa to nitrogen is mainly regulated by the urea cycle and TCA cycle, which provides a reference basis for breeders to select and develop quinoa strains that are tolerant to low- or high-nitrogen stress in the future, and lays a theoretical foundation for improving the nitrogen fertilizer utilization efficiency of quinoa and other plants.

## 4. Materials and Methods

### 4.1. Materials and Treatment

Advanced generation cultivars of red and white quinoa (Dianli-1299 and Dianli-71) were planted in the Scientific Research Base of Yunnan Agricultural University in Xundian County, Kunming (102°41′ E, 25°20′ N). This area is characterized by a subtropical plateau monsoon climate. In this study, red and white quinoa are represented as R and W, respectively. An equal number of red and white quinoa seeds were selected and evenly planted in a pot. Three gradients were set for the amount of CH_4_N_2_O applied; LN represents 0 kg/hm^2^ of CH_4_N_2_O (urea) representing the nitrogen-deficient treatment (includes LN-R and LN-W), CK represents 112.5 kg/hm^2^ of CH_4_N_2_O representing the normal-nitrogen treatment (includes CK-R and CK-W), and HN represents 337.5 kg/hm^2^ of CH_4_N_2_O representing the high-nitrogen treatment (includes HN-R and HN-W). During the experiment, the amount of P_2_O_5_ fertilizer used was 112.5 kg/hm^2^, and the amount of K_2_O fertilizer used was 112.5 kg/hm^2^. In the early stages, conventional cultivation and management techniques were adopted (average temperature: 25.6 °C, sunshine duration: approximately 10 h, sowing depth: 2–3 cm; substrate: contents of CH_4_N_2_O, P_2_O_5_, and K_2_O were 2.75 g/kg, 1.66 g/kg, and 1.18 g/kg, respectively). Fertilization was conducted at the two-leaf stage of quinoa. The growth of the quinoa peaked after 30 d of fertilization treatment; thus, this was considered the best sampling time. We simultaneously collected samples from all treatments on the same day, which was characterized by the following environmental conditions. Three samples of seedling leaves were then placed in liquid nitrogen for freezing and transferred to −80 °C immediately upon collection for storage. The metabolomic and transcriptomic analyses were then conducted on the leaves of the quinoa seedlings, with 3 biological replicates and 18 samples in total (Wuhan MetWare Biotechnology Co. Ltd., Wuhan, China; https://www.metware.cn).

### 4.2. Morphological Data Collection

Leaf height and area of the quinoa seedlings were determined in triplicate after 30 d of fertilization treatment. A Vernier caliper was used to measure the height of each leaf. The height was measured as the distance from the base to the top of a stretched leaf. A TPYX-A crop leaf morphometry was used to determine the area of the individual leaves (Zhejiang, China, https://www.tpyn.net, accessed on 18 April 2021).

### 4.3. Metabolite Extraction and Qualitative and Quantitative Analysis

Quinoa seedling leaf samples were then placed in a freeze-drying machine (Scientz-100F; Ningbo Scientz Biotechnology Co. Ltd., Zhejiang, China) for vacuum freeze-drying after 30 d of fertilization treatment. Thereafter, the samples were powdered using a grinder (MM400; Retsch GmbH, Haan, Germany) and then dissolved in methanol. The supernatant was clarified using eddy current centrifugation (12,000 rpm, 10 min, 4 °C), and analyzed through ultrahigh-performance liquid chromatography–tandem mass spectrometry (UPLC-MS/MS) analysis. The data acquisition instrument system included ultrahigh-performance liquid chromatography (UPLC) (SHIMADZU Nexera X2; https://www.shimadzu.com.cn/, accessed on 15 June 2021) and tandem mass spectrometry (MS/MS) (Applied Biosystems 4500 QTRAP; http://www.appliedbiosystems.com.cn/, accessed on 15 June 2021) and an MS database (MWDB; http://en.metware.cn/list/27.html, accessed on 15 June 2021). The qualitative analysis of the sample was then performed through secondary mass spectrometry (MS). During the analysis, the isotopic signal, specifically, the repeated signal containing K+, Na+, and NH4+, and the repeated signal of the fragment ion itself (i.e., which represents other substances with a high molecular weight) were removed. A multiple reaction monitoring (MRM) mode of triple quadrupole MS was used to quantify metabolites, obtain different metabolite spectra, integrate peak areas, and conduct integration correction [61]. During the instrument-based analyses, to monitor the repeatability of the analysis process, a quality control sample was run for every 10 samples. Multivariate statistics were then used to retain the original information, reduce the dimensionality of the data, establish a reliable mathematical model, and use the built-in statistical prcomp function in the R software (www.r-project.org/, accessed on 15 June 2021) to visualize the differences among sample groups [62]. The Pheatmap package in R was then used to draw a heat map, and hierarchical cluster analyses were conducted to analyze the accumulation of metabolites across the different samples. Additionally, orthogonal partial least squares discriminant analysis (OPLS-DA) was then used to extract components from the independent variable X and dependent variable Y and subsequently screen for variables underlying the differences among the samples [63,64]. The variable importance in projection (VIP) of the multivariate analysis of the OPLS-DA model was obtained, and DEMs were further screened for *p*-value and fold change [65]. The identified metabolites were then annotated using the KEGG compound database (http://www.kegg.jp/kegg/compound/, accessed on 15 June 2021), and the annotated metabolites were mapped to the KEGG pathway database (http://www.kegg.jp/kegg/pathway.html, accessed on 15 June 2021) [66]. The pathways with notable regulation of metabolites were input into the metabolite set enrichment analysis, and their significance was determined through a hypergeometric test.

### 4.4. Transcriptome Sequencing and Data Analysis

The sequencing of the transcriptome occurred in the following three stages: RNA extraction and detection, library construction and quality control, and sequencing based on the Illumina HiSeq platform, which was conducted by Beijing Novozyme Technology Co., Ltd. (www.noZvogene.com.cn, accessed on 15 June 2021). The integrity of RNA was first determined through agarose gel electrophoresis; thereafter, a Qubit 2.0 Fluorometer (Thermo Fisher Scientific, Waltham, MA, USA) and Agilent 2100 bioanalyzer (Agilent Technologies, Santa Clara, CA, USA) were used to detect the concentration and integrity of RNA prior the construction of a library, and therefore the library was built only after the quality of the RNA was ensured. The starting RNA amount used for library preparation was the total RNA extracted from a sample, and it was ≥1 µg in all samples. Illumina’s NEBNext UltraTM RNA Library Prep Kit (Illumina, San Diego, CA, USA) was used to prepare the library. After the libraries were constructed, they were initially quantified using a Qubit 2.0 Fluorometer and then diluted to 1.5 ng/uL, and the insert size of the libraries was confirmed using an Agilent 2100 bioanalyzer. The effective concentration of the libraries was accurately quantified by qRT-PCR after the insert size had met expectations. After a certain insert size was reached, qRT-PCR was performed to accurately quantify the effective concentration of the library to ensure that it was of high quality. After the libraries were confirmed to be of high quality, the different libraries were pooled according to the target offline data volume and sequenced using the Illumina platform, and 150 bp paired-end reads were generated. Fastpv0.19.3 was used to filter the offline data. Reads with adapter sequences, nitrogen content exceeding 10% of the alkali bases of the read, or bases of low quality (Q ≤ 20) alkali exceeding 50% of the read were removed; both paired reads were removed in the latter two cases to obtain clean data. HISAT v2.1.0 [67] was used to build an index, compare clean reads with the specified reference genome, and obtain mapped data. StringTie v1.3.4 [68] was used to predict new genes, and FeatureCounts v1.6.2 [69] was used to calculate the gene alignment. Differential expression analysis between groups was performed using DESeq2 v1.22.1 [70], and *p* values were corrected using the Benjamini–Hochberg method. Thereafter, the FPKM of the genes was calculated according to their respective lengths as an indicator of transcript or gene expression level. Functional annotation of differentially expressed genes was performed using the Kyoto Encyclopedia of Genes and Genomes (KEGG), Gene Ontology (GO), Karyotic Orthologous Groups (KOG), PfAM, Swiss-Prot, TrEMBL, and NR databases. Pathway significant enrichment analysis was performed to identify pathways that were significantly enriched in differentially expressed genes compared to the whole genomic background by performing hypergeometric distribution tests in the KEGG database in terms of pathways.

### 4.5. Combined Transcriptome and Metabolome Analyses

According to the DEMs combined with results from the analysis of DEGs, the DEMs and the DEGs in the same processing were simultaneously mapped to the KEGG pathway map. The purpose of this was to elucidate relationships between the genes and metabolites. Based on the enrichment analysis results of DEMs and DEGs, bar graphs were drawn to show the differences in the enrichment of metabolites and gene pathways. A correlation analysis was conducted for genes and metabolites detected in each treatment; specifically, a Pearson correlation coefficient of genes and metabolites was calculated using the COR package in R. Additionally, a correlation analysis was conducted for DEGs and DEMs, and the results associated with Pearson correlation coefficients greater than 0.8 were extracted and used to draw a correlation coefficient clustering heat map [71]. Canonical correlation analysis (CCA) was used to process DEGs and DEMs to establish an OPLS-DA model; this analytical approach was also used to preliminarily determine the variables associated with high correlations and weights across different sample groups through a loading diagram [72].

### 4.6. RT-qPCR

RNA samples that were used for real-time quantitative polymerase chain reaction (RT-qPCR) were extracted from the leaves of quinoa seedlings of Dianli-1299 and Dianli-71 cultivars. To verify the reliability of the transcriptome sequencing results, all samples of genes on pathways related to nitrogen metabolism were selected for subsequent RT-qPCR experiments. The reagent used in this experiment was PerfectStartTM SYBR qPCR Supermix (TransGen Biotechnology, Beijing, China). RT-qPCR was performed using a 96-well plate on the StepOnePlus instrument (Applied Biosystems, CA, USA). The PCR was performed in three repeats, and the results were compared with the internal reference values for TUB-6. The relative transcription level was calculated using the 2^−ΔΔCt^ method [73].

## 5. Conclusions

We conducted a metabolomic and transcriptomic analysis of quinoa seedlings under nitrogen-deficient and high-nitrogen conditions. The phenotypic characteristics of quinoa seedlings were greatly affected by varying concentrations of nitrogen. We found that quinoa seedlings responded to these treatments through the regulation of metabolites and some genes involved in carbon fixation in photogenic organisms, arginine biosynthesis, and alanine, aspartate, and glutamate metabolism pathways. Additionally, the different nitrogen treatments had varying effects on DEGs and DEMs of quinoa seedlings. We found that 15 DEMs and 8 DEGs were the key factors that resulted in the differences observed in quinoa seedlings under different nitrogen conditions. Quinoa seedlings were able to cope in different nitrogen fertilizer environments by primarily regulating arginine biosynthesis. Our study provides a reference for breeders to select and develop quinoa cultivars that are resistant to low- or high-nitrogen stress, to facilitate a deeper understanding of the metabolic processes under varying nitrogen concentrations. This study also provides a theoretical foundation for a science-driven application of nitrogen fertilizer and an improvement of the NUE of quinoa.

## Figures and Tables

**Figure 1 ijms-24-11580-f001:**
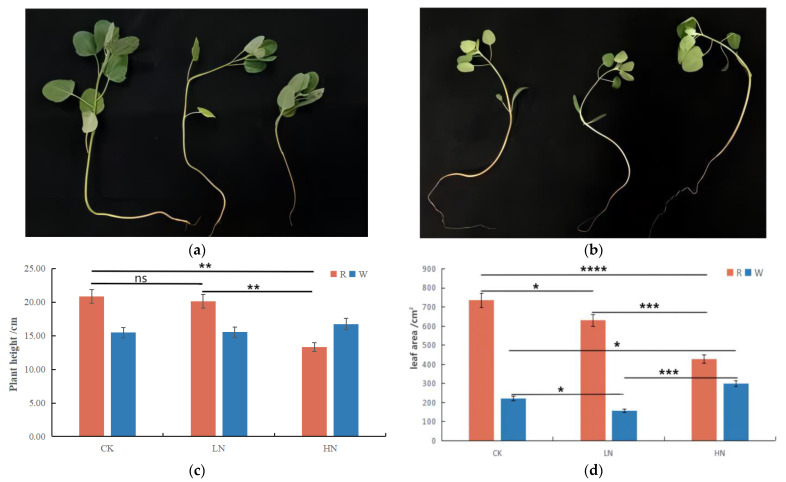
(**a**,**b**) Quinoa seedlings with different nitrogen fertilizer dosages. From left to right, the growth of red quinoa and white quinoa seedlings in CK, LN, and HN groups is compared. (**c**) The height of quinoa cultivars in CK, LN, and HN groups at the seedling stage. R and W represent red quinoa and white quinoa, respectively. All differences in plant height between white quinoa are not significant. (**d**) Leaf area of quinoa cultivars in CK, LN, and HN groups at the seedling stage. CK, control; LN, low nitrogen; HN, high nitrogen. R and W represent red quinoa and white quinoa, respectively. “*****” represents *p* ≤ 0.05, “******” represents *p* ≤ 0.01, “*******” represents *p* ≤ 0.001, “********” represents P ≤ 0.0001, “ns” represents *p* > 0.05.

**Figure 2 ijms-24-11580-f002:**
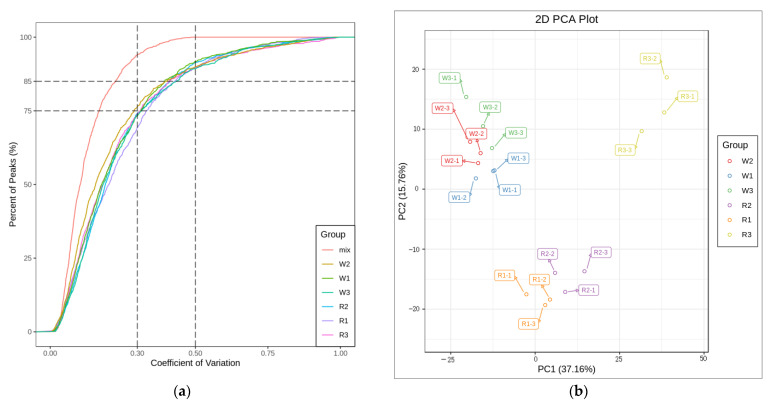
(**a**) Coefficient of variation (CV) distribution. (**b**) Principal component analysis (PCA) score chart. (**c**) Venn diagram of DEMs (CK vs. LN and CK vs. HN). (**d**) Overall cluster analysis heatmap of samples. R and W represent red quinoa and white quinoa, respectively; 1, 2, and 3 represent LN, CK, and HN, respectively.

**Figure 3 ijms-24-11580-f003:**
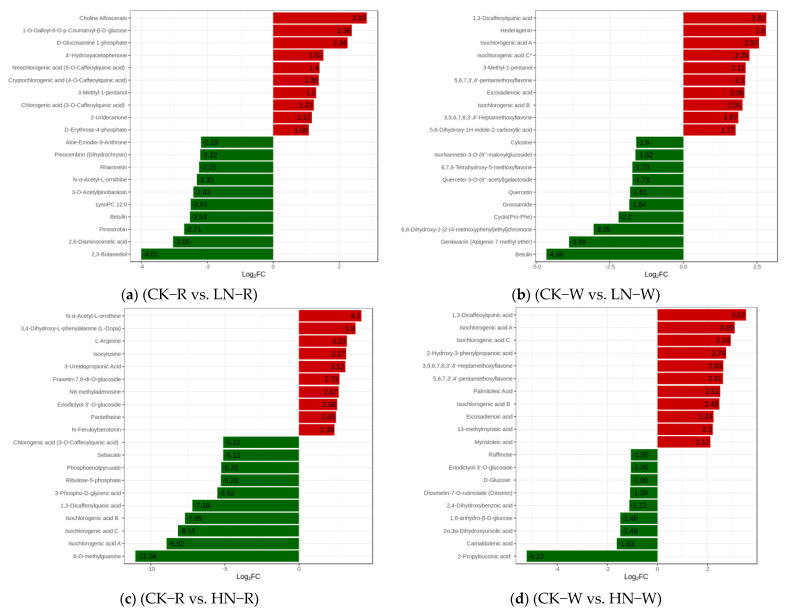
(**a**–**f**) Difference multiple bar chart of DEMs. (**g**) K−means diagram of DEMs. Log2FC and DEMs in the abscissa and ordinates in the histogram of DEM abundance are shown. The X−coordinate represents the name of the sample, the Y−coordinate represents the relative content of the standardized metabolite. R and W represent red quinoa and white quinoa, respectively; 1, 2, and 3 represent LN, CK, and HN, respectively.

**Figure 4 ijms-24-11580-f004:**
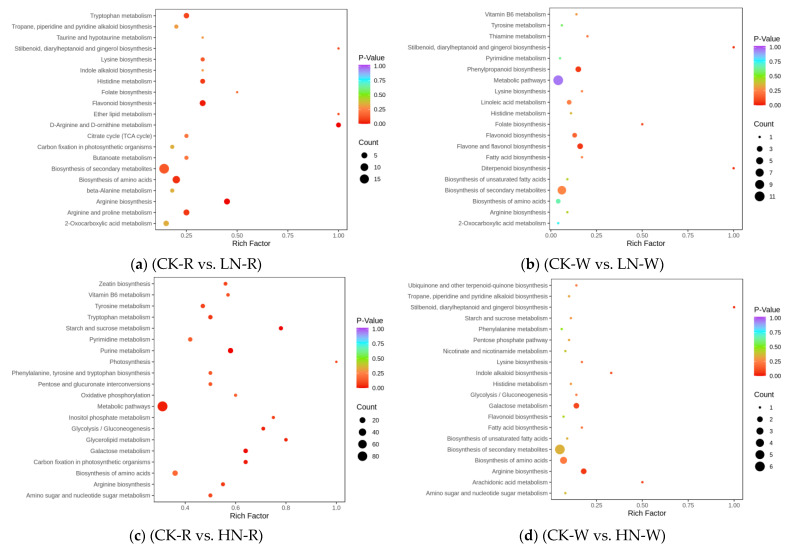
(**a**–**f**) KEGG enrichment graph for DEMs; (**a**,**c**,**e**) LN vs. CK, HN vs. CK, and LN vs. HN for red quinoa, respectively; (**b**,**d**,**f**) LN vs. CK, HN vs. CK, and LN vs. HN for white quinoa, respectively. The abscissa represents the Rich Factor corresponding to each path, and the ordinate represents the path name. The color of the point reflects the *p*-value size. The redder the point is, the more significant the enrichment. The size of the dot represents the number of enriched differential metabolites.

**Figure 5 ijms-24-11580-f005:**
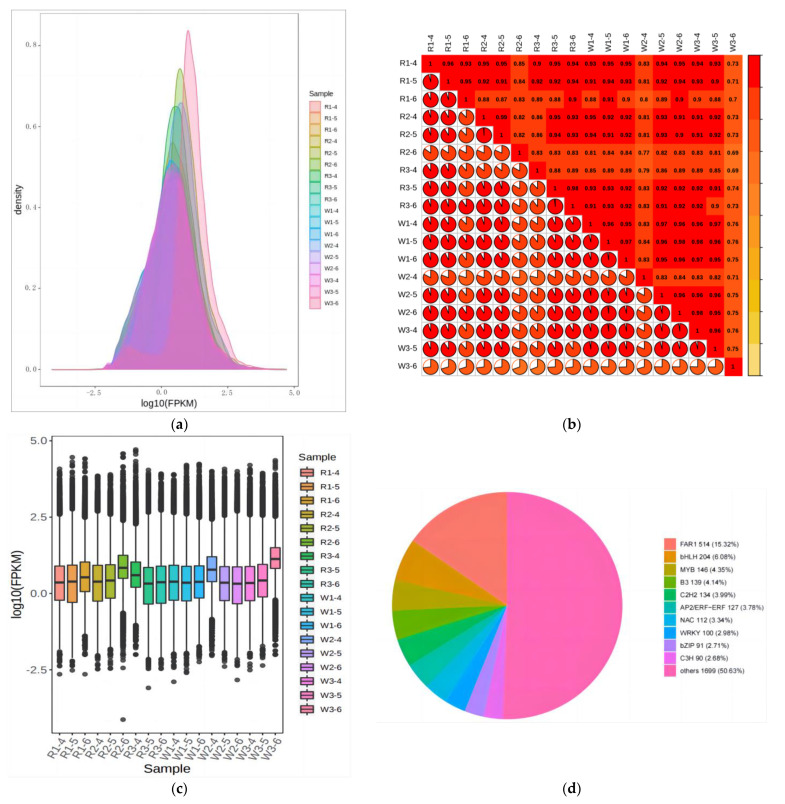
(**a**) Expression density distribution plot. (**b**) Correlation heat map. (**c**) Gene expression box line plot. (**d**) Percentage diagram of transcription factors. R and W represent red quinoa and white quinoa, respectively; 1, 2, and 3 represent LN, CK, and HN, respectively.

**Figure 6 ijms-24-11580-f006:**
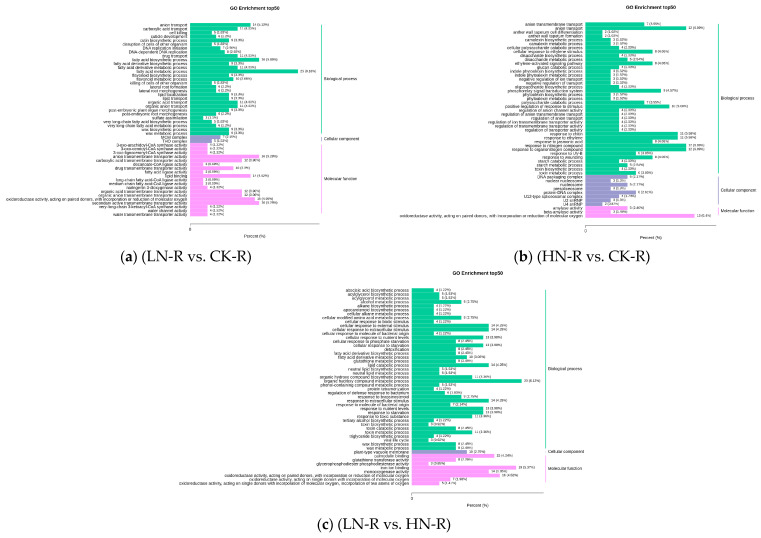
(**a**–**c**) DEG enrichment analysis in Gene Ontology (GO). The abscissa represents the ratio of the genes annotated to the entry to the whole quantity of annotated genes, and the ordinate represents the title of the GO entry.

**Figure 7 ijms-24-11580-f007:**
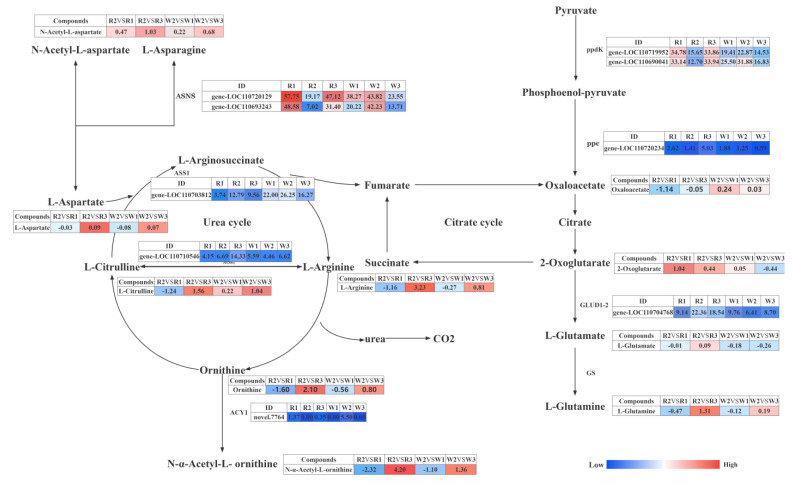
Response mechanism of carbon fixation in photogenic organisms, arginine biosynthesis, alanine, aspartate, and glutamine metabolism pathways to nitrogen treatment in quinoa seedlings. The gene expression level was expressed through the FPKM value, while the metabolite level was expressed through the log2FC value of different control groups. ASS1, argininosuccinate synthase [EC:6.3.4.5]; ACY1, aminoacylase [EC:3.5.1.14]; NOS1, nitric oxide synthase [EC:1.14.13.39]; ASNS, asparagine synthase (glutamine-hydrolyzing) [EC:6.3.5.4]; ppdK, pyruvate, orthophosphate dikinase [EC:2.7.9.1]; ppc, phosphoenolpyruvate carboxylase [EC:4.1.1.31]; GLUD1_2, glutamate dehydrogenase [EC:1.4.1.3]. The boxes in the pathway represent differentially expressed genes or differentially expressed metabolites. Blue represents downregulated genes or metabolites, whereas red represents upregulated genes or metabolites. R and W represent red quinoa and white quinoa, respectively; 1, 2, and 3 represent LN, CK, and HN, respectively.

**Figure 8 ijms-24-11580-f008:**
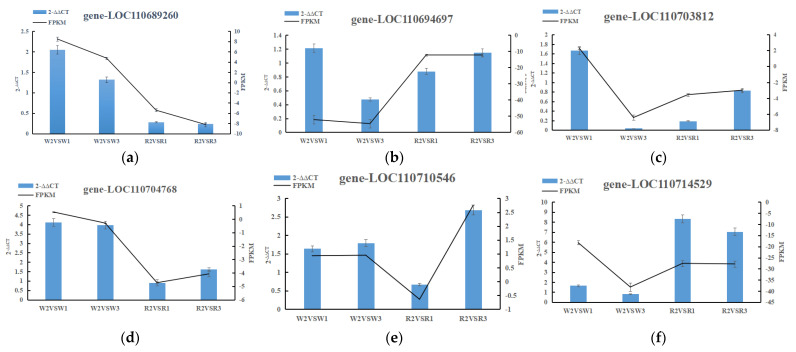
Validation of the transcription levels for the selected differentially expressed genes via RT-qPCR.

**Table 1 ijms-24-11580-t001:** Correlation analysis of DEMs and DEGs.

Group	Gene-ID	EC	Metaname	Formula	Compound	PCC
CK vs. LN	gene-LOC110704768	glutamate dehydrogenase	pme2380	C5H6O5	α-Ketoglutaric acid	−0.851
CK vs. HN	gene-LOC110710546	nitric oxide synthase	pme2527	C5H12N2O2	L-Ornithine	0.84
	gene-LOC110710546	nitric oxide synthase	pme0193	C5H10N2O3	L-Glutamine	0.809
	gene-LOC110710546	nitric oxide synthase	mws0260	C6H14N4O2	L-Arginine	0.864
	gene-LOC110710546	nitric oxide synthase	Zmyn000155	C7H14N2O3	N-α-Acetyl-L-ornithine	0.800

Note: EC represents enzyme digestion sites in related pathways; PCC represents Pearson’s correlation coefficient.

**Table 2 ijms-24-11580-t002:** Primer sequences for validating genes.

Quantity	Gene-ID	NCBI-Gene ID	Primer	5′ to 3′
1	gene-LOC110689260	110689260	Forward Primer	CTCTGACTATGATTGAACA
			Reverse Primer	CAATAGCAACCAAGAATG
2	gene-LOC110694697	110694697	Forward Primer	TACTTCTCATACCCTATCA
			Reverse Primer	CATCAACTTCTCACTGTA
3	gene-LOC110703812	110703812	Forward Primer	CTTAATCCTGCTCTCAAT
			Reverse Primer	ATTCCTGTCTCTGCTATA
4	gene-LOC110704768	110704768	Forward Primer	GGTGTTATCATTCTTCCT
			Reverse Primer	TCTTCAGTTCATTGTTCA
5	gene-LOC110710546	110710546	Forward Primer	CTATATCTGACGCTCTTG
			Reverse Primer	TCACATATTCACTCTCATC
6	gene-LOC110714529	110714529	Forward Primer	GAATCCTCCATCTTACAG
			Reverse Primer	CTTATCTTATGCTCTTCCA
Internal reference gene	TUB-6	831100	Forward Primer	TGAGAACGCAGATGAGTGTATG
			Reverse Primer	GAAACGAAGACAGCAAGTGACA

## Data Availability

The datasets presented in this study can be found in online repositories. The names of the repository/repositories and accession number(s) are as follows: https://www.ncbi.nlm.nih.gov/ (accessed on 15 May 2023), PRJNA946047; the number of SRA is processing.

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
