# Peer review of "Transcriptome and Metabolome Analyses Reveal Mechanisms Underlying the Response of Quinoa Seedlings to Nitrogen Fertilizers"

_ijms, 2023, doi:10.3390/ijms241411580_

Round 1
Reviewer 1 Report
Dear Authors,
You combined both metabolomic and transcriptomic assays on two quinoa variants in response to different amounts of nitrogen to assess the NUE. I believe your research can be greatly appreciated by researchers who are interested in aggregating metabolomics and transcriptomics. The results also can be interesting to those who are concerned with the excessive usage of nitrogen from an environmental perspective. Your presentation and the methodology are acceptable; however, I think the paper can be improved after covering/addressing some minor drawbacks.
Here are my main concerns about the manuscript.
- I wonder why you haven’t compared LN and HN treatments in most of your analysis. Is that something you did, and it didn’t go well? OR any other specific reason?
- some of the figures in the manuscript lack self-descriptive captions making the plots more complicated. They need to be self-explanatory.
- Haven’t you applied TMM normalization in DESeq2? If you have, what was the point of using FPKM before which is not recommended in most RNA-Seq analyses?
- I also got a bit confused about how you named the treatments in different sections of the manuscript. I think a more general naming system would help readers to interpret the experiment much easier (e.g., LN-R-1).
You may also find the file enclosed containing detailed corrections and suggestions. I left some corrections/suggestions regarding the above-mentioned concerns.
I hope these suggestions can have a minor positive impact on improving your valuable work.
Cheers,
Reviewer

See the comments in the review report file (the PDF)
Author Response
Dear Reviewer:
Thank you for your constructive feedback on our manuscript entitled “Transcriptome and Metabolome Analyses Reveal Mechanisms Underlying the Response of Quinoa Seedlings to Nitrogen Fertilizers” (ijms-2444774). Our revised submission is attached.
I am very sorry that have parts was not clear in the original manuscript. We would like to thank your valuable suggestions. We have accepted all suggestions and made appropriate modifications. Each of your questions was answered below. The manuscript was also revised and revisions were marked in yellow highlight.
We look forward to hearing from you regarding our submission. We would be glad to respond to any further questions and comments that you may have. best wishes!
Point 1: I wonder why you haven’t compared LN and HN treatments in most of your analysis. Is that something you did, and it didn’t go well? OR any other specific reason?
Response 1: I am so sorry that have parts was not clear in the original manuscript. The comparison between LN and HN has been added to the corresponding analysis. (Line171-172,194-200,204-205,212-213,260-262,265-267,281-282)
Point 2: some of the figures in the manuscript lack self-descriptive captions making the plots more complicated. They need to be self-explanatory.
Response 2: I am so sorry that have parts was not clear in the original manuscript. I have rechecked the title and made some modifications. The definition of R,Wand numbers has been added in the caption. (Figure2,Figure3,Figure5,Figure7)
Point 3: Haven’t you applied TMM normalization in DESeq2? If you have, what was the point of using FPKM before which is not recommended in most RNA-Seq analyses?
Response 3: I'm sorry for not expressing myself clearly, which led you to mistakenly believe that we are using FPKM for difference analysis. We are using Deseq2, and the input for difference analysis is the count value, not fpkm; When making differences in DESEQ2, built-in functions are used to normalize the input data; TMM is a commonly used data normalization method when using edgeR for difference analysis, and Deseq2 does not frequently use this method.
Point 4: I also got a bit confused about how you named the treatments in different sections of the manuscript. I think a more general naming system would help readers to interpret the experiment much easier (e.g., LN-R-1).
Response 4: I am so sorry that have parts was not clear in the original manuscript. The naming has been changed according to a more general naming system.(Figure3,4,6)

Reviewer 2 Report
In the current manuscript, the authors have conducted multi-omics analysis (transcriptome and metabolome) to understand how quinoa seedlings respond to nitrogen fertilizers. The study falls well within the scope, and the work is well-comprehensive. However, I have some concerns and comments that need to be addressed before acceptance.
1. Figures: Very low quality. Seems to be a screenshot. Please modify.
2. Figure 1c: The statistics presented in the current form might be complicated for the readers.
3. Figure 1d: Please correct the spelling of “area” in the vertical axis.
4. It is not clear why the authors chose red and white quinoa seedlings.
5. Figures 2, 3, etc.: It is not clear what are SR, R, etc. Please mention the meaning of these abbreviated forms in the figure legend.
6. “Transcriptome sequencing and data analysis”: please mention the platform (for eg: Novaseq or HiSeq).
7. Many important references are missing, such as HISAT v2.1.0, StringTie v1.3.4d, DESeq2 v1.22.1, etc.
8. I recommend rewriting the “Discussion” section. In the current form, the authors have discussed way more about previous results. I recommend focusing more on the obtained data and how they can be used in the future for this species as well as for related species.
Moderate editing of the English language is required.
Author Response
Dear Reviewer:
Thank you for your constructive feedback on our manuscript entitled “Transcriptome and Metabolome Analyses Reveal Mechanisms Underlying the Response of Quinoa Seedlings to Nitrogen Fertilizers” (ijms-2444774). Our revised submission is attached.
I am very sorry that have parts was not clear in the original manuscript. We would like to thank your valuable suggestions. We have accepted all suggestions and made appropriate modifications. Each of your questions was answered below. The manuscript was also revised and revisions were marked in yellow highlight.
We look forward to hearing from you regarding our submission. We would be glad to respond to any further questions and comments that you may have. best wishes!
Point 1: Figures: Very low quality. Seems to be a screenshot. Please modify.
Response 1: I am so sorry that have parts was not clear in the original manuscript. The quality of the figures in the article has been modifed.
Point 2: Figure 1c: The statistics presented in the current form might be complicated for the readers.
Response 2: I am so sorry that have parts was not clear in the original manuscript. The difference in plant height between different nitrogen fertilizer levels of white quinoa is "ns". To make it easier for readers to read, it has been deleted from the figure and reflected in the annotations.
Point 3: Figure 1d: Please correct the spelling of “area” in the vertical axis.
Response 3: I'm sorry for this error. The spelling of 'area' has been modified
Point 4: It is not clear why the authors chose red and white quinoa seedlings.
Response 4: I am so sorry that have parts was not clear in the original manuscript. In our previous work, we conducted the same treatment on 30 quinoa high generation lines independently selected by Yunnan Agricultural University in four colors of red, white, yellow and black, and their performance was basically the same. The two most representative varieties were Chenopodium album (Dianli-1299) and Chenopodium album (Dianli-71); And the seedling stage is a transitional stage from heterotrophic to autotrophic, which is a key period to ensure high crop yield. If the growth status of the seedling stage is restricted, it will have a negative impact on yield(line77-79). Therefore, we selected the seedlings of these two strains for subsequent experiments.
Point 5: Figures 2, 3, etc.: It is not clear what are SR, R, etc. Please mention the meaning of these abbreviated forms in the figure legend.
Response 5: I am so sorry that have parts was not clear in the original manuscript. R represents red quinoa, and there is no "SR" in the original text. For example, "R1VSR2" is "R1 vs. R2", which has been modified.
Point 6: “Transcriptome sequencing and data analysis”: please mention the platform (for eg: Novaseq or HiSeq).
Response 6: I am so sorry that have parts was not clear in the original manuscript. As mentioned in “Transcriptome sequencing and data analysis”, sequencing using Illumina HiSeq platform.(line 545)
Point 7: Many important references are missing, such as HISAT v2.1.0, StringTie v1.3.4d, DESeq2 v1.22.1, etc.
Response 7: I am so sorry that have parts was not clear in the original manuscript. The missing references have been cited. (References67-70)
Point 8: I recommend rewriting the “Discussion” section. In the current form, the authors have discussed way more about previous results. I recommend focusing more on the obtained data and how they can be used in the future for this species as well as for related species.
Response 8: I am so sorry that have parts was not clear in the original manuscript. Modifications have been made in the discussion section, and the obtained data has been presented and future prospects have been made. (line368-375,399-401,405-413,417-424,427-431,436-440,468-472)

Round 2
Reviewer 2 Report
Dear authors,
The manuscript has been substantially improved now. I recommend it for publication. However, please do the following minor amendments before acceptance:
1. Please check the gene names throughout the manuscript. They must be in italics.
2. Please, some of the figures are not yet of publication standard. Please revise them. For example figure 8.
Wish you good luck,
Regards,
Reviewer
Minor editing of the English language is required.